# Reliable *N*-Glycan Analysis–Removal of Frequently Occurring Oligosaccharide Impurities by Enzymatic Degradation

**DOI:** 10.3390/molecules28041843

**Published:** 2023-02-15

**Authors:** Robert Burock, Samanta Cajic, René Hennig, Falk F. R. Buettner, Udo Reichl, Erdmann Rapp

**Affiliations:** 1MPI for Dynamics of Complex Technical Systems, Sandtorstraße 1, 39106 Magdeburg, Germany; 2glyXera GmbH, Brenneckestraße 20, 39120 Magdeburg, Germany; 3Institute of Clinical Biochemistry, Hannover Medical School, Carl-Neuberg-Straße 1, 30625 Hannover, Germany; 4Bioprocess Engineering, Otto-von-Guericke University, Universitätsplatz 2, 39106 Magdeburg, Germany

**Keywords:** glucosidase, dextranase, glucoamylase, glycoprotein, removal, oligosaccharide ladder, polyhexose contamination, xCGE-LIF, stem cell lysate

## Abstract

Glycosylation, especially *N*-glycosylation, is one of the most common protein modifications, with immense importance at the molecular, cellular, and organismal level. Thus, accurate and reliable *N*-glycan analysis is essential in many areas of pharmaceutical and food industry, medicine, and science. However, due to the complexity of the cellular glycosylation process, in-depth glycoanalysis is still a highly challenging endeavor. Contamination of samples with oligosaccharide impurities (OSIs), typically linear glucose homo-oligomers, can cause further complications. Due to their physicochemical similarity to *N*-glycans, OSIs produce potentially overlapping signals, which can remain unnoticed. If recognized, suspected OSI signals are usually excluded in data evaluation. However, in both cases, interpretation of results can be impaired. Alternatively, sample preparation can be repeated to include an OSI removal step from samples. However, this significantly increases sample amount, time, and effort necessary. To overcome these issues, we investigated the option to enzymatically degrade and thereby remove interfering OSIs as a final sample preparation step. Therefore, we screened ten commercially available enzymes concerning their potential to efficiently degrade maltodextrins and dextrans as most frequently found OSIs. Of these enzymes, only dextranase from *Chaetomium erraticum* and glucoamylase P from *Hormoconis resinae* enabled a degradation of OSIs within only 30 min that is free of side reactions with *N*-glycans. Finally, we applied the straightforward enzymatic degradation of OSIs to *N*-glycan samples derived from different standard glycoproteins and various stem cell lysates.

## 1. Introduction

*N*-glycans play an important role as one of the most frequently occurring post-translational modifications of eukaryotic proteins. They influence, for example, the structure, stability, and binding properties of secreted and membrane-bound proteins. Therefore, numerous investigations focus on *N*-glycans to identify biomarkers, e.g., of diseases [1,2,3,4,5,6] or stem cells [7,8,9,10]. Furthermore, *N*-glycan analysis is part of biopharmaceutical protein characterization, e.g., as a critical quality parameter of therapeutic monoclonal antibodies [11,12] or their biosimilarity assessment [13,14]. For these reasons, reliable *N*-glycan analysis should provide unambiguous identification and accurate quantification of glycan species even in complex biological samples. A frequently occurring problem in glycan analysis is the contamination of samples with free oligosaccharides [15,16,17,18,19,20,21,22,23,24]. Based on our own unpublished results and rare literature identifications [25,26,27,28], maltodextrins with linear α(1-4)-linkages and dextrans with mainly α(1-6)-linkages are common oligosaccharide impurities (OSIs, also called polyhexose contaminations) in *N*-glycan samples. Such OSIs can complicate analyses based on mass spectrometry [28,29,30], liquid chromatography [8,25,31] and capillary electrophoresis (CE) [27,32].

For analysis by these methods, glycans are often derivatized at their reducing end to facilitate their mass spectrometric and optical detectability and/or their separation [33]. A reducing end is a common feature of many carbohydrates, including OSIs, which will be concurrently labeled with *N*-glycans. As the molecular characterization of glycan analytes relies on physicochemical properties (hydrophilicity, hydrophobicity, ionic interactions, electromobility or molecular mass [34]), the similarity of *N*-glycans and OSIs in these properties often causes co-detection (overlapping signals) of both [26,35,36,37]. If samples are contaminated, OSIs might be identified by a characteristic ladder-like signal pattern. However, in some cases, they do not show such a pattern. Thus, they might remain unnoticed. If recognized, it is often not possible to distinguish plain OSI signals, plain *N*-glycan signals, and overlapping signals caused by both without further investigation. Unfortunately, complete separation of OSIs from *N*-glycans after *N*-glycan release is very difficult. Hence, up to now, the removal of OSIs is based on time- and labor-intensive additional sample preparations on glycoprotein level [5,29,36,38,39]. This may require additional sample material, which is often limited. Due to these disadvantages, it is common practice to exclude assumed OSI signals during data evaluation [7,8,9,32,40,41,42]. However, excluding these signals carries the risk of also excluding overlapping signals caused by *N*-glycans. Due to all these drawbacks of common strategies that compromise reliable glycoanalysis, we have developed an alternative approach to deal with OSIs in *N*-glycan samples.

Not all OSIs interfere with the analysis of *N*-glycans to the same extent. Problems in data evaluation are caused only by those OSIs that are detected at the same mass-to-charge ratio, and in the same retention or migration time ranges like *N*-glycans. An important influence on these ranges is the molecular size of oligosaccharides. *N*-glycans share a common trimannosyl chitobiose core structure [43]—a pentasaccharide that is further extended by addition of different sugars. This common core structure can be considered as a lower molecular size limit of *N*-glycans. Thus, especially OSIs with a similar size, specifically with a degree of polymerization of 4 (DP4) or larger, might interfere with *N*-glycan analysis. Based on this premise, we developed an efficient method for the enzymatic degradation of interfering OSIs (DP4 and larger) to non-interfering OSIs (DP3 and smaller) inside *N*-glycan samples that are already fully processed for analysis. By doing so, already-prepared sample material can be used, which minimizes working time and the risk of excluding *N*-glycan signals during data evaluation.

Promising enzyme candidates for this task are glucoside hydrolases. Therefore, we screened a selection of ten commercially available glucoside hydrolases using multiplexed capillary gel electrophoresis with laser-induced fluorescence detection (xCGE-LIF) [22,44,45,46,47,48,49,50,51,52] with 8-aminopyrene-1,3,6-trisulfonic acid (APTS)-labeled maltodextrins and dextrans as standard substrates. We also paid particular attention to show the activity of selected enzymes on OSIs in a complex sample matrix, while having no side-activity on *N*-glycans. For this reason, we tested the selected enzymes on APTS-labeled *N*-glycans derived from the standard glycoproteins bovine immunoglobulin G (IgG), fetuin and ribonuclease B, and on artificially OSI-contaminated *N*-glycan samples of these proteins. Furthermore, we compared our fast and easy approach of enzymatic degradation for removal of OSIs with commonly applied strategies for OSI removal on the glycoprotein level, such as filtration, immobilization on a polyvinylidene difluoride (PVDF) membrane or in a gel block. Finally, we demonstrated the great potential of our method on biologically relevant *N*-glycan samples derived from various stem cell lysates with suspected OSIs.

## 2. Results

### 2.1. Depletion of OSIs on the Glycoprotein Level

Sample contamination with OSIs is a frequently appearing and known problem in glycan analysis. Accordingly, we also observed OSIs in many instances, including single glycoproteins such as erythropoietin or human IgG, and complex biological specimens such as basophils or zebra fish embryos (Appendix A). As a very prominent example, the xCGE-LIF analysis of *N*-glycans derived from human embryonic stem cells (hESCs) [10] revealed a ladder-like series of highly abundant peaks (Figure 1A). Based on migration time matching to the maltodextrin peak pattern, we could assign these peaks to this common type of OSIs (Figure 1B). Obviously, the OSIs present in the hESC-derived sample prevent proper *N*-glycan analysis and must be removed. Therefore, this sample was used for testing and evaluating the efficiency of non-enzymatic approaches to remove OSIs directly from glycoprotein-containing samples before *N*-glycan release. Hence, (glycol-) proteins were retained on a 10 kDa Amicon Ultra-0.5 centrifugal filter (Figure 1C), embedded into a polyacrylamide gel block (Figure 1D) or immobilized on a PVDF membrane (Figure 1E) and subsequently extensively washed with buffer to remove OSIs. While these treatments considerably increased the detectability of *N*-glycans, the removal of OSIs was incomplete for all tested approaches. Our findings underline the necessity to develop an alternative approach for a reliable and complete removal of interfering OSIs.

### 2.2. Enzymatic Degradation of APTS-Labeled Maltodextrins and Dextrans

As all tested non-enzymatic approaches failed to sufficiently remove OSIs from stem cell-derived glycoprotein samples, we considered enzymatic approaches using glucoside hydrolases for degradation of interfering OSIs to non-interfering (small) OSIs. Therefore, we initially analyzed maltodextrins and dextrans by xCGE-LIF and observed that OSIs with DP3 and DP4 migrate at approximately 110 MTU` and 150 MTU’, respectively (Figure 2A,B). Of note, even highly sialylated *N*-glycans are typically detected at migration times above 130 MTU` in xCGE-LIF, as exemplarily shown for bovine fetuin-derived *N*-glycans (Appendix A). Thus, we defined OSIs with DP3 and smaller as non-interfering OSIs. As they do not affect *N*-glycan analysis, they can remain as reaction products after treatment with the glucoside hydrolases.

Subsequently, we systematically evaluated the capability of different glucoside hydrolases to degrade OSIs, making use of the potent high throughput capabilities of xCGE-LIF. We screened the enzyme performances under varying reaction conditions, specifically by using different reaction buffer compositions at different pH values, and various incubation times, as described in detail in Section 4.4. As substrates, we tested both APTS-labeled maltodextrins and dextrans to evaluate the activity and linkage preference of the applied enzymes. The degradation efficiency was assessed by determining individual relative peak heights of desired, non-interfering reaction products (DP1, DP2, and DP3) as well as the sum of relative peak heights of all interfering OSIs (≥DP4). For short incubation times, we observed that OSI degradation was the most efficient in WS0095 (Appendix A) and in disodium phosphate-citrate buffer, both at pH 5 (Table 1). In other buffers tested, we could not observe a sufficient degradation of interfering to non-interfering OSIs within a short incubation time, as exemplarily shown for oligo-α-(1,4-1,6)-glucosidase treatment in sodium phosphate-citrate buffer at pH 7 (Appendix A). The slightly better degradation of dextrans by dextranase from *Chaetomium erraticum* (DxChe) in the disodium phosphate-citrate buffer at pH 5 (60.3% DP2 and 29.8% DP3 compared to 47.3% DP2 and 44.7% DP3 in WS0095) led to the selection of this buffer for further experiments.

APTS-labeled maltodextrins were not degraded by the oligo-α-(1,4-1,6)-glucosidase and only very slightly by the oligo-1,6-glucosidase (Table 1, fourth and last row). All other glucoside hydrolases digested maltodextrins to varying extents, showing different specificities and efficiencies. For example, the α-glucosidase barely degraded larger maltodextrins (>DP4; shown in Appendix A) even upon incubation times >1 h (data not shown) but produced quite effectively monosaccharides (46%) from maltodextrins with DP2 to DP4. In contrast, other enzymes could effectively degrade maltodextrins >DP4, but produced mainly maltodextrins with DP2 and DP3 and no monosaccharides. Second-best results were obtained with β-amylase (Table 1, highlighted in light blue). After 1 h incubation, mainly maltodextrins with DP2 (57.4%) were produced, and only 6.5% remained with DP4 or higher. Best results were obtained with glucoamylase P (GAP, Table 1, highlighted in dark blue). The enzyme degraded maltodextrins completely to DP2 and DP3, and no higher DPs were detected anymore. Further experiments showed that the desired degradation of maltodextrins to maltose (DP2) and maltotriose (DP3) was already achieved after 30 min of incubation, even at 10-fold lower GAP concentration (Figure 2A,C).

APTS-labeled dextrans were degraded exclusively by GAP, oligo-1,6-glucosidase, dextranase from *Penicillium* species (DxPsp), and DxChe. While a 1 h treatment by GAP and oligo-1,6-glucosidase reduced the dextrans ≥DP4 from 98.8% to 36.5% and 48.0% (Table 1, highlighted light green), the DxChe and DxPsp treatment reduced dextrans ≥DP4 to less than 10% (highlighted dark green). This result could also be achieved with DxChe already after 30 min incubation (Figure 2B,D).

Based on their remarkable performance in degrading OSIs, GAP, β-amylase, DxChe and DxPsp were chosen for further evaluation of possible side reactions with APTS-labeled *N*-glycans.

### 2.3. Assessment of Side Reactions with APTS-Labeled N-Glycans

Unwanted side reactions of the chosen enzymes were tested on *N*-glycans derived from the bovine glycoproteins IgG, ribonuclease B, and fetuin (annotated *N*-glycan fingerprints are provided in the Appendix A) [45,51]. These glycoproteins contain a wide range of different *N*-glycan structures, including *N*-glycans with terminal galactose, *N*-acetylglucosamine, mannose, fucose and sialic acid. The chosen glucoside hydrolases could be therefore tested for side activities such as β-galactosidase, β-*N*-acetylglucosaminidase, α-mannosidase, α-fucosidase and α-neuraminidase (sialidase). Overnight treatment of APTS-labeled *N*-glycans with DxChe or GAP showed no detectable unwanted side activity on *N*-glycans (see Appendix A). Accordingly, a 30 min treatment of samples using these two enzymes resulted in unimpaired, highly reproducible normalized peak heights (Appendix A). In contrast, β-galactosidase and *N*-acetylglucosaminidase activities were observed for DxPsp (see Appendix A), as well as for β-amylase as shown in Figure 3 (and Appendix A). Additionally, β-amylase showed α-mannosidase activity (Appendix A). Thus, due to their side reactions, β-amylase and DxPsp are unsuitable for the degradation of interfering OSIs inside *N*-glycan samples and were consequently excluded from further experiments. In addition, the purity on the protein level of all enzyme formulations was examined using Sodium dodecyl sulfate -polyacrylamide gel electrophoresis (SDS-PAGE) as described before [53]. Besides the protein bands corresponding to the calculated molecular weight of the respective glucoside hydrolase (marked with arrows), we found several protein bands not corresponding to the respective glucoside hydrolase in nearly all enzyme preparations (Appendix A).

### 2.4. Enzymatic Degradation of OSIs in N-Glycan Samples

Based on our previously described findings on enzymatic activities and specificities, we decided to test GAP and DxChe for their potential to degrade interfering maltodextrins or dextrans, respectively, directly in *N*-glycan samples. Therefore, APTS-labeled maltodextrins or dextrans were spiked into APTS-labeled *N*-glycans derived from bovine ribonuclease B, fetuin, and IgG. Each enzyme was able to degrade the respective OSI type in a complex sample containing *N*-glycans without side reactions. In Figure 4, this is exemplarily shown for interfering maltodextrins (comigrating with *N*-glycans) in a bovine ribonuclease B *N*-glycan sample. Apparently, the APTS-labeled *N*-glycan peak marked with a black arrow at approximately 250 MTU’ is overlapping with the peak of APTS-labeled maltodextrin with DP7 (Figure 4B). Based on migration time matching only, this peak and several OSI peaks might be mistakenly assigned as *N*-glycans. Thus, the relative quantification of actual *N*-glycans is impaired. The interfering maltodextrins (comigrating with *N*-glycans) were successfully degraded through the enzymatic treatment to non-interfering OSIs (<DP4), and the bovine ribonuclease B-derived *N*-glycan fingerprint was restored unimpaired (Figure 4C).

As the ultimate proof-of-concept, we evaluated the efficiency of enzymatic degradation of OSIs in complex *N*-glycan mixtures derived from human-induced pluripotent stem cells (hiPSCs) and hiPSCs differentiated towards cardiomyocytes (hiPSC-CMs). As described previously [10,38], the differentiation of hiPSCs into hiPSC-CMs causes major alterations in xCGE-LIF-derived fingerprints (Figure 5A). Based on migration time matching, we suspected some of the strikingly deviating peaks to potentially arise from maltodextrins. For this reason, we treated the APTS-labeled hiPSC- and hiPSC-CM-derived *N*-glycans with GAP, as it has been shown to be superior to DxChe for the degradation of maltodextrins in this study. Treatment with GAP completely degraded the interfering maltodextrin-based OSIs to non-interfering OSIs inside our hiPSC- and hiPSC-CM-derived *N*-glycan samples within 30 min at 37 °C (Figure 5B,C).

GAP was further tested for its ability to degrade OSIs in other complex *N*-glycan samples, i.e., derived from zebra fish embryos (Appendix A), basophils (Appendix A), and derived from cultured human and murine embryonic stem cells (hESCs and mESCs, Appendix A). The two pluripotent cell lines display common, but also some species-specific *N*-glycans (e.g., Neu5Ac vs. Neu5Gc, presence of *N*-acetylgalactosamine and α-galactose in murine cells) [51]. Like for the hiPSCs-derived sample (Figure 5B, black graph), a typical and easy-to-recognize ladder pattern was observed in the hESC-derived sample (Appendix A, black graph). In contrast, the mESCs-derived sample (Appendix A, black graph) contained fewer OSIs without such a pronounced and easily recognizable pattern, similarly to the hiPSC-CMs-derived sample (Figure 5C, black graph). Regardless of the amount and pattern, OSIs interfering with the *N*-glycan analysis of these samples were successfully degraded with GAP (Appendix A, red graphs).

## 3. Discussion

Oligosaccharides are ubiquitous and OSIs are frequently observed during analysis of glycan samples [15,16,17,18,19,20,21,22,23,24]. Accordingly, we detected OSIs mostly comprising maltodextrins and dextrans by xCGE-LIF-based *N*-glycan analysis, too. These OSIs often interfered with *N*-glycan peaks, complicating, or even prohibiting proper *N*-glycan peak quantification. Varying amounts and DP-ranges of these OSIs [8,18,20,21,30] often make it difficult to recognize the typical ladder-like pattern (Figure 5B, Appendix A) [18,36,39,41]. The unnoticed OSIs could be misclassified as *N*-glycans, leading to structural misinterpretations.

As shown, affected sample types range from single glycoproteins such as erythropoietin [55] and human IgG to complex biological specimen such as basophil granulocytes [56], zebra fish embryos and stem cell lysates (Figure 1 and Figure 5, and Appendix A). In the literature, only in a few instances could the occurrence of OSIs be attributed to the biological background. For example, degrading glycogen from liver-derived tissue samples [25,27,29], cell wall polysaccharides from plant-derived [57], or fungal-derived expression systems [58] were assigned to OSIs. More often, OSIs are introduced from external sources [23], which is also in line with our experience. Possible sources that might contaminate samples with OSIs during sample collection and preparation are powdered gloves [59], cell culture media [8,58,60], chromatographic material [39], or inadequately pure chemicals and consumables (Appendix A). Ideally, contamination of samples by OSIs through these sources should be avoided, but often, the precise origin of OSIs remains unknown. Identifying the exact source of contamination requires time-consuming and expensive testing, which often does not give a definitive answer. Unfortunately, even when identified, the origin of OSIs is often difficult to avoid, especially if OSIs arise from the cell culture medium [8,58,60] or cultivation feed [35,36].

Therefore, previous strategies aimed to remove contaminations such as OSIs already on the glycoprotein level by additional preparative sample clean-up steps such as: filtration, sodium dodecyl sulfate polyacrylamide gel electrophoresis (SDS-PAGE) [38,53,59], capturing of glycoproteins using PVDF membranes [61], extraction or precipitation [57], and dialysis [29]. Following these strategies, we could show that OSIs could not be completely removed from glycoprotein samples (see Figure 1), and sometimes, OSIs are introduced by these sample preparation steps (Appendix A). This finding is in accordance with, e.g., Aoki et al. [36], who state only a reduction in OSI signals by washing precipitated glycoproteins with ice-cold acetone, or Jeong et al. [29], who still detected OSI signals after dialysis of glycoproteins. Additionally, analysis of large numbers of samples in high throughput might be hampered due to the introduction of additional, labor- and time-intensive purification steps. To overcome these problems, we developed an enzymatical approach to degrade OSIs inside the fully processed (released and labeled) *N*-glycan sample. This additional step enables a repeated xCGE-LIF analysis, without the need to again perform a time- and sample-consuming, adapted sample preparation.

The use of a glucoside hydrolase-supported degradation of OSIs is rarely reported in the literature, especially regarding the use in *N*- or *O-*glycan analysis, and no comprehensive characterization of the used enzymes or detailed description of the procedure is available [25,27,28]. For this reason, we here provide a thorough examination of glucoside hydrolases for the purpose of OSI degradation within *N-*glycan samples. Because of its high-throughput capabilities, xCGE-LIF is predestined for screening purposes. Therefore, we monitored the outcome of the glucoside hydrolase treatments of APTS-labeled maltodextrins, dextrans, and *N*-glycans using this analytical method.

First, the most effective glucoside hydrolases were identified regarding reaction time and completeness in several buffers, differing, e.g., in pH or composition. After comparing the tested buffers for each enzyme, the pH turned out to be the most relevant factor. Overall, we observed a slower degradation of OSIs in less acidic buffers, even when the buffer matches the recommended pH of the respective enzyme. In the more acidic buffers WS0095 and disodium phosphate-citrate buffer (both at pH 5) some enzymes readily produced high amounts of non-interfering OSIs even after short incubation times. With only one exception, the results obtained for the four most effective enzymes (DxChe, DxPsp, GAP, and β-Amylase) were very similar between these two buffers. DxChe showed a higher amount of dextran dimer (DP2) after 1 h in the disodium phosphate-citrate buffer at pH 5 (Table 1) compared to WS0095 (Appendix A). Therefore, we decided to use the disodium phosphate-citrate buffer at pH 5 for the subsequent experiments and will focus on the results of OSI degradation in this buffer.

Maltodextrins with a wide DP range were degraded best by GAP and β-amylase (Table 1). The different ratios of reaction products obtained by GAP and β-amylase can be explained by the enzymatic reaction mechanisms. GAP (EC 3.2.1.3) [62] is an exoglucosidase that releases β-D-glucose by cleaving terminal linkages at the non-reducing end of α(1-4) linked oligo- and polysaccharides. Thus, it can digest maltotriose (DP3), but only acts poorly on maltose (DP2), which is in accordance with observations by Fagerström [63]. The complete degradation of interfering to non-interfering maltodextrins with GAP, (Figure 2C), is in good accordance with results of Hanzawa et al. [25], who unfortunately provided no information on the reaction conditions. In contrast to GAP, the β-amylase cleaves the second α(1-4)-glucosidic linkage at the non-reducing end of maltodextrins and releases two glucose units at a time (i.e., maltose), making it an exopolysaccharidase [64]. Accordingly, we observed an accumulation of maltose (DP2) and maltotriose (DP3) during the β-amylase treatment of maltodextrins (Table 1), but no considerable formation of glucose (DP1) from maltotriose. This is in accordance with published observations by Morell et al. [65] and Yoshigi et al. [66], who describe maltotriose as poor substrate for β-amylase.

The selective degradation of small maltodextrins by the α-glucosidase is a further interesting observation. A preference for smaller oligosaccharides compared to polysaccharides is described for α-glucosidases (EC number 3.2.1.20) in the ENZYME Database [62]. However, since the size of oligosaccharides is usually defined up to DP12 [67], it is surprising that the tested α-glucosidase (with 1 U/μL) barely digested maltodextrins with DP5 or higher (Appendix A). Donczo et al. identified maltotriose and maltotetraose in various mouse tissue samples upon overnight incubation with an excessive amount of α-glucosidase (100 U) [27], but did not explicitly comment on larger maltodextrins. In view of our results, it remains uncertain whether the amount of the α-glucosidase used by the authors is sufficient to degrade large maltodextrins.

Dextrans were best degraded by DxPsp and DxChe. In general, dextrans are branched polymers, but are often produced for biotechnological purposes by *Leuconostoc mesenteroides* as highly linear polysaccharides [68,69]. These polysaccharides contain over 95% α(1-6)-linkages and approximately 5% α(1-3)-linkages that may introduce branching points in the molecule. Upon short incubation times, the two dextranases degraded more than 90% of dextrans ≥DP4 into smaller oligosaccharides ≤DP3 but failed to completely degrade all dextrans ≥DP4 (Table 1). Both dextranases are classified under the EC number 3.2.1.11 [62] as endoglucosidases, cleaving α(1-6)-linkages. For this reason, some remaining oligomers are likely to contain α(1-3)-linkages at or in proximity to preferred cleavage sites towards the reducing end, inhibiting a complete degradation of these dextran-based OSIs. Therefore, complete degradation of all interfering dextran-based OSIs to non-interfering OSIs is not achievable by the tested dextranases. Nevertheless, they are good options to identify dextran based OSIs.

Of all tested glycoside hydrolases, GAP and β-amylase degraded most efficiently maltodextrins, as well as DxPsp and DxChe dextrans. As a next step, these enzymes were thoroughly tested to exclude side reactions with APTS-labeled *N*-glycans, which would impair the final analysis of the sample’s *N*-glycome. The enzymes β-amylase and DxPsp showed arbitrary side reactions with *N*-glycans (Figure 3 and Appendix A), making these enzymes unsuitable for the degradation of interfering OSIs in *N*-glycan samples. To evaluate the origin of the side activities, we checked the purity of the enzymes with SDS-PAGE (Appendix A) and found several protein bands in the β-amylase (Lanes 2 and 3) and at least one additional protein band in DxPsp (Lanes 4 and 5). For this reason, it remains unclear whether the observed side reactions originate from the tested enzymes themself, or from an exoglycosidase contamination. As additional protein bands are also visible for several other enzymes, we would rather call the tested glucoside hydrolases enzyme preparations. These results clearly suggest that utmost care must be taken when selecting and using enzymes for reliable glycoanalytics. Furthermore, reducing the enzyme concentration to a suitable minimum also decreases the risk of side reactions. Hence, we also tested lower enzyme concentrations, and we could reduce the used concentration of GAP for the final application in complex samples.

We also applied our enzymatic approach on biologically relevant, OSI-contaminated *N*-glycan samples derived from zebra fish embryos (Appendix A), basophils (Appendix A), as well as pluripotent human and murine stem cells (Figure 5 and Appendix A). Thereby, we could show the wide applicability and robustness of applying the two remaining enzymes, GAP and DxChe. They reliably degraded all interfering OSIs to non-interfering OSIs under varying conditions, i.e., buffers, incubation times, concentrations, and in different complex samples. By showcasing the wide occurrence in different sample types, we underlined the importance of OSI degradation for accurate *N*-glycan analysis. Just as one example, pluripotent stem cells have the amazing capability to differentiate into nearly all somatic cell types. Because of this property, stem cells, including hESCs and hiPSCs, have become useful tools in research and provide great promise in regenerative medicine [70]. Accurate *N*-glycan profiling of pluripotent stem cells and their derivatives is of utmost importance; e.g., glycan-based markers are invaluable tools for characterization and isolation of these cells.

In particular, the typical ladder pattern of OSIs is not always obvious in *N*-glycan fingerprints of, e.g., the hiPSC-CM-derived sample (Figure 5). Not recognizing the presence of the OSIs in this sample could lead to structural misinterpretation and thus to incorrect assignment of a biomarker for this type of differentiated stem cells. Such OSIs that are difficult to detect from the signal pattern occur frequently. Therefore, samples should always be checked for the presence of OSIs. Applying the carefully selected glucoside hydrolases GAP or DxChe, we are now able to detect and degrade the interfering OSIs inside *N*-glycan samples without altering the *N*-glycan fingerprints (Figure 4 and Figure 5, and Appendix A). This makes our enzymatic approach a valuable option for inclusion to routine workflows, enabling reliable *N*-glycan analysis by ensuring identification and exclusion of OSIs signal. The treatment of contaminated samples with GAP or DxChe for 30 min at 37 °C facilitates the fast degradation of interfering OSIs in high throughput, leaves APTS-labeled *N*-glycans unimpaired, and enables remeasurement of the sample after a single working day.

## 4. Material and Methods

### 4.1. Materials

Ultrapure water with a conductivity of 1 μS was produced in-lab by a MilliQ^®^ Reference Water Purification System A^+^ from Merck KGaA (Darmstadt, Germany).

Acetonitrile (ACN) was purchased from VWR (Darmstadt, Germany).

Amicon Ultra 0.5 mL centrifugal filters with an exclusion cutoff of 10 kDa and Immobilon-PSQ PVDF membrane were obtained from Millipore (Darmstadt, Germany).

Bovine ribonuclease B, fetuin and IgG, dextran MW5000, maltooligosaccharides from corn sirup (maltodextrins), phosphate-buffered saline (PBS), potassium phosphate monobasic, citric acid monohydrate, α-amylase II-A, α-amylase IX-A, α-amylase XIII-A, α-glucosidase, β-amylase II-B, and dextranases from *Penicillium* species (DxPsp) and *Chaetomium erraticum* (DxChe) were purchased from Sigma-Aldrich (Hamburg, Germany).

The sample preparation kit glyXprepCE™ for *N*-glycan release, APTS-labeling, and post labeling sample cleanup was from glyXera (Magdeburg, Germany).

SDS was purchased from Applichem (Darmstadt, Germany).

Glucoamylase P (GAP), oligo-α-1,6-glucosidase, and oligo-α-(1,4-1,6)-glucosidase were obtained from Megazyme (Wicklow, Ireland).

WS0049 buffer (final concentration 50 mM disodium phosphate, pH 6.6), WS0095 buffer (final concentration 100 mM sodium acetate, 2 mM zinc chloride, pH 5), and WS0122 buffer (final concentration 100 mM disodium phosphate-citrate, 50 μg/mL BSA, pH 6) were acquired from Prozyme Inc. (Hayward, CA, USA).

Disodium phosphate dihydrate was from Riedel-de Haën (Seelze, Germany).

Disodium phosphate-citrate buffers at pH 5 and 7 [71] and potassium phosphate buffer (150 mM, acidified to pH 6 with phosphoric acid) were prepared as 10X stock solutions.

Diverse stem cell lysates were used to explore the efficiencies of OSI depletion and enzymatic degradation in a complex sample. Human embryonic stem cells (hESCs), human-induced pluripotent stem cells (hiPSCs), and hiPSC-derived cardiomyocytes (hiPSC-CM) for glycan analysis were obtained as described before [10,38]. Murine embryonic stem cells (mESCs) for glycan analysis were obtained as described by Abeln et al. [51].

Basophil samples were kindly provided by Dr. Olaf Rötzschke [56] from Singapore Immunology Network (SIgN), A*STAR (Agency for Science, Technology and Research), Singapore.

Zebra fish embryos were a generous gift of Dr. Philipp Schmalhorst from the Institute of Science and Technology Austria (IST Austria), Klosterneuburg, Austria.

Erythropoietin was kindly provided by Matthias Meininger [55], Max Planck Institute for Dynamics of Complex Technical Systems, Magdeburg, Germany.

### 4.2. Non-Enzymatic Approaches for Depletion of OSIs on the Glycoprotein Level

Non-enzymatic approaches are based on washing out the OSIs from retained or immobilized glycoproteins. Three different methods were applied to the lysates of hESCs comprising immobilization of glycoproteins (i) on a 10 kDa Amicon Ultra-0.5 centrifugal filter, (ii) in a gel block, and (iii) on a PVDF membrane. The immobilization of glycoproteins within a gel block or on a PVDF membrane, as well as the subsequent *N*-glycan release, were performed as described in [61,72]. For the 10 kDa Amicon Ultra-0.5 centrifugal filter approach, hESCs resolved in 2% SDS solution were applied to the filter unit and washed three times with PBS. Subsequent enzymatic release of *N*-glycans from retrieved glycoproteins was performed as described in Section 4.3. Fluorescent labeling of released *N*-glycans, depletion of the excess label, as well as xCGE-LIF analysis were performed as described in Section 4.3.

### 4.3. xCGE-LIF Analysis of N-Glycans and Oligosaccharides

To evaluate the performance of non-enzymatic approaches for OSI depletion and the progress of enzyme reactions, glyXboxCE™ (glyXera, Magdeburg, Germany) was used, including the sample preparation kit glyXprepCE™. Briefly, bovine ribonuclease B, fetuin, and IgG as standard glycoproteins and stem cell lysates were dissolved in PBS and de-*N*-glycosylated by peptide-*N*-glycosidase F according to the kit instructions. Maltodextrins and dextrans were dissolved in ultrapure water. Maltodextrins, dextrans, and released *N*-glycans were labeled with APTS and purified by hydrophilic interaction chromatography (HILIC) in solid phase extraction (SPE) mode as described in the kit instructions. APTS-labeled analytes were analyzed with xCGE-LIF using glyXboxCE™. Obtained data was processed using glyXtoolCE™ (glyXera GmbH, Magdeburg, Germany) [73]. The software performed a migration time alignment of electropherograms to an internal standard, resulting in aligned *N*-glycan fingerprints. Peaks with a signal-to-noise ratio >10 were picked, and relative peak intensities were calculated based on peak height normalization. *N*-glycan structures were determined with the glyXtoolCE™ integrated *N*-glycan database and confirmed with exoglycosidase digests (see Appendix A).

### 4.4. Enzymatic Degradation of APTS-Labeled Maltodextrins and Dextrans

Ten glucoside hydrolases were screened regarding their ability to digest APTS-labeled maltodextrins and dextrans under different reaction conditions, i.e., buffers. Tested glucoside hydrolases were: α-amylase II-A (0.15 U/μL), α-amylase IX-A (1 U/μL), α-amylase XIII-A (1 U/μL), α-glucosidase (1 U/μL), β-amylase (1 U/μL), oligo-1,6-glucosidase (1 U/μL), oligo-α-(1,4-1,6)-glucosidase (1 U/μL), GAP (0.17 U/μL and 0.017 U/μL), DxPsp (0.5 U/μL), and DxChe (activity unknown). Overall, the pH ranges and optima for the tested enzymes’ activities varied from acidic (pH 3) to neutral (pH 6.9) conditions according to the manufacturer’s information. To exclude the loss of acid labile sialic acids from *N*-glycans, we tested six digestion buffers in less acidic to neutral range: WS0049 (pH 6.6), WS0095 (pH 5), WS0122 (pH 6), disodium phosphate-citrate buffer at pH 5 and 7, and potassium phosphate buffer at pH 6.

The reaction setup was as follows: All ten glucoside hydrolases were tested in time series from 10 min up to overnight incubation. For each sampling time point, APTS-labeled maltodextrins or dextrans were formulated in 9 μL 1X digestion buffer. To each sampling time point, 1 μL of glucoside hydrolase solution was added and thoroughly mixed. The reaction mixture was incubated at 37 °C for 10 min, 30 min, 1 h, 4 h, or overnight (18 h). The reaction was stopped by the addition of 90 μL 89% ACN_aq_ (*v*/*v*) and drying in a vacuum centrifuge. Samples were subsequently stored at −20 °C and formulated in Washing Solution I (part of the glyXprepCE™ kit) for sample purification. As a negative control, 9 μL of buffered APTS-labeled sample were treated similar without addition of glucoside hydrolase solution.

To deplete the enzymes and salts from the reaction mixture before xCGE-LIF analysis, an adapted sample purification using the glyXprepCE™ kit was performed. After application of the APTS-labeled sample, washing was performed four times with Washing Solution I before elution. The result of the enzyme reaction was monitored by xCGE-LIF on a glyXboxCE™ system.

### 4.5. Test for Side Reactions with APTS-Labeled N-Glycans

Possible side reactions of selected enzymes (GAP, DxChe, DxPsp and β-amylase) with *N*-glycans were evaluated using APTS-labeled *N*-glycans derived from bovine standard glycoproteins ribonuclease B, fetuin, and IgG. Specifically, activity on galactose, *N*-acetylglucosamine, mannose, fucose and sialic acid residues was monitored. All enzyme reactions were performed as described in Section 4.4. using disodium phosphate-citrate buffer at pH 5 as the digestion buffer.

### 4.6. Enzymatic Degradation of OSIs in Complex N-Glycan Samples

APTS-labeled *N*-glycans from standard glycoproteins (see Section 4.5) spiked with APTS-labeled maltodextrins or dextrans, and several stem-cell-derived *N*-glycan samples containing OSIs, were treated with GAP or DxChe as described in Section 4.4. using disodium phosphate-citrate buffer at pH 5 for 30 min.

## 5. Concluding Remarks

We observed that OSIs are a frequent contamination in various samples and represent an underestimated problem for accurate and reliable glycan analysis. Due to similar physicochemical properties, signals of *N*-glycans and OSIs might overlay, which impedes or even prevents proper *N*-glycan analysis. Furthermore, typical ladder patterns are not always observed, and OSIs varying in size and amounts might remain unrecognized. Therefore, we performed a thorough examination of different glucoside hydrolases and established a procedure for the enzyme-mediated degradation of OSIs interfering with the analysis of *N*-glycans: The treatment of already labeled *N*-glycan samples with suitable glucoside hydrolases—which is advantageous to usually performed glycoprotein sample clean-up steps that are not completely removing all OSIs during sample preparation. Therefore, one should consider integrating the herein-described fast and easy enzymatic treatment into regular sample preparation procedures. This straightforward approach for selective enzymatic degradation to remove OSIs is expected to become an invaluable tool for obtaining a true glycan picture from complex samples.

## Figures and Tables

**Figure 1 molecules-28-01843-f001:**
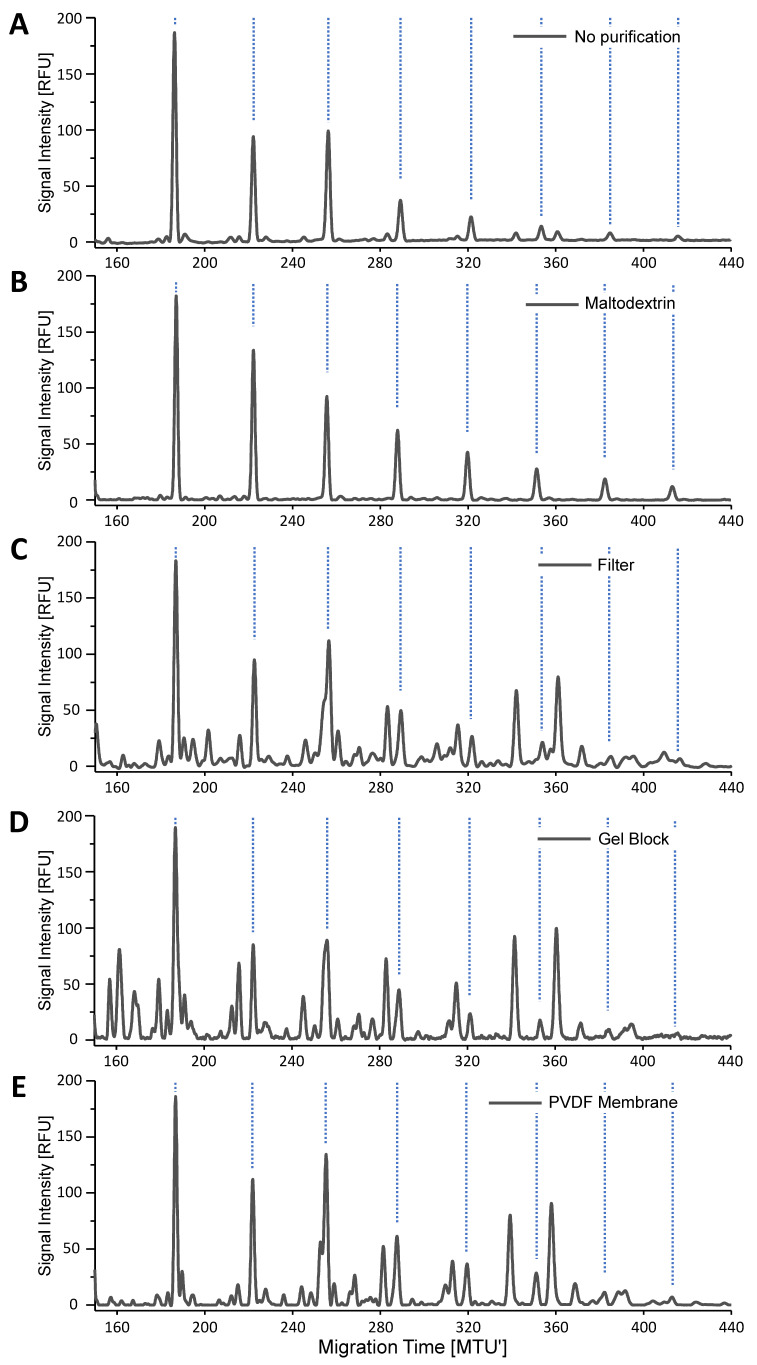
xCGE-LIF analysis of APTS-labeled *N*-glycans from human embryonic stem cells (hESCs) without purification (**A**) and a maltodextrin standard to identify signals related to OSIs (**B**). The hESCs were purified on the (glyco-) protein level prior to *N*-glycan release by the following methods: filtration using 10 kDa molecular weight cut-off filter (**C**), incorporation of sample into gel block (**D**), and immobilization of glycoproteins on PVDF membrane (**E**). *X*-axis was aligned to an internal migration time standard, resulting in Migration Time Units (MTU’). Signal intensity is displayed in relative fluorescence units (RFU). Data analysis was performed using glyXtoolCE™.

**Figure 2 molecules-28-01843-f002:**
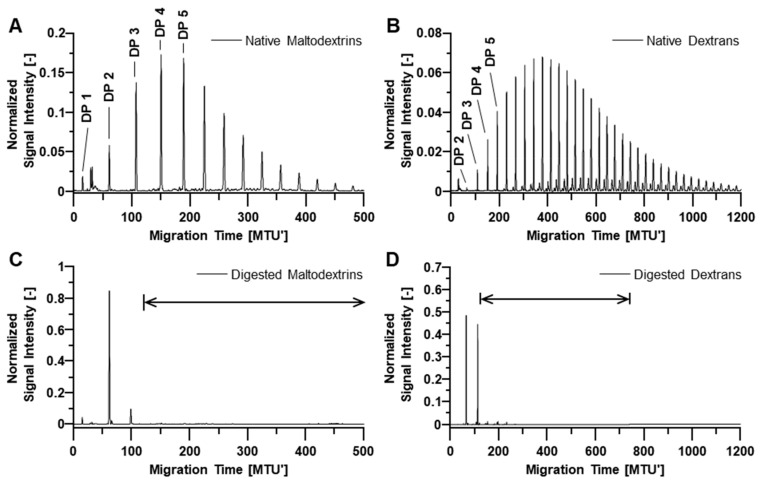
APTS-labeled maltodextrins (**A**) and dextrans (**B**) analyzed by xCGE-LIF before and after enzymatic treatment with glucoside hydrolases. Maltodextrins were degraded by glucoamylase P (GAP) with 0.017 U/μL (**C**). Dextrans were degraded by dextranase from *Chaetomium erraticum* (DxChe, **Panel D**). APTS-labeled maltodextrins and dextrans were formulated in disodium phosphate-citrate buffer at pH 5 for enzymatic degradation. Interfering oligosaccharides (Degree of Polymerization DP ≥ 4) could be successfully degraded to non-interfering oligosaccharides (DP1, DP2 and DP3) within an incubation time of 30 min. Normalized signal intensity (with sum of picked peak heights = 1) was plotted over aligned migration time in Migration Time Units (MTU’). Maltodextrin and dextran peaks are exemplarily annotated up to DP5 in (**A**,**B**). The migration time range of *N*-glycans is indicated by an arrow in (**C**,**D**). Data analysis was performed using glyXtoolCE™.

**Figure 3 molecules-28-01843-f003:**
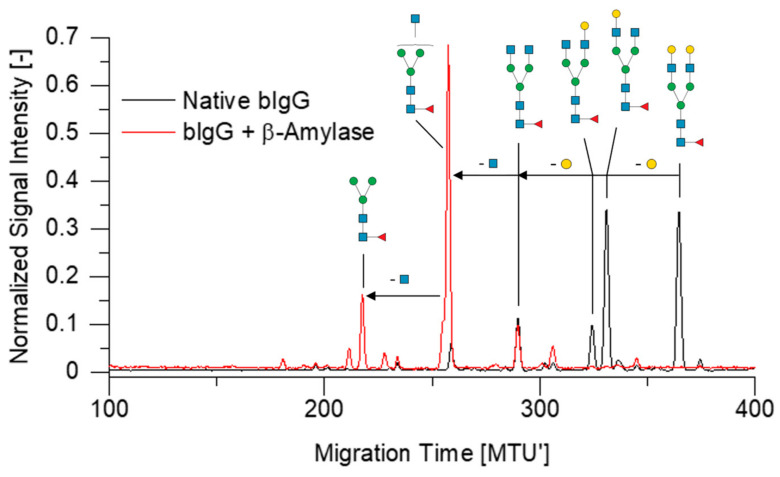
Overlay of xCGE-LIF generated fingerprints of bovine IgG (bIgG)-derived *N*-glycans before and after overnight treatment with β-amylase. β-galactosidase and *N*-acetylglucosaminidase side reactivity of β-amylase were revealed. Migration time plotted on the *x*-axis was aligned to an internal migration time standard, resulting in Migration Time Units (MTU’). Signal intensity (peak height) of automatically picked peaks was normalized to a sum value of 1. Annotation of *N*-glycan peaks is based on earlier reports [45]. Data analysis was performed using glyXtoolCE™. Symbolic representation of *N*-glycan structures follows the guidelines of Symbol Nomenclature for Glycans (SNFG) [54].

**Figure 4 molecules-28-01843-f004:**
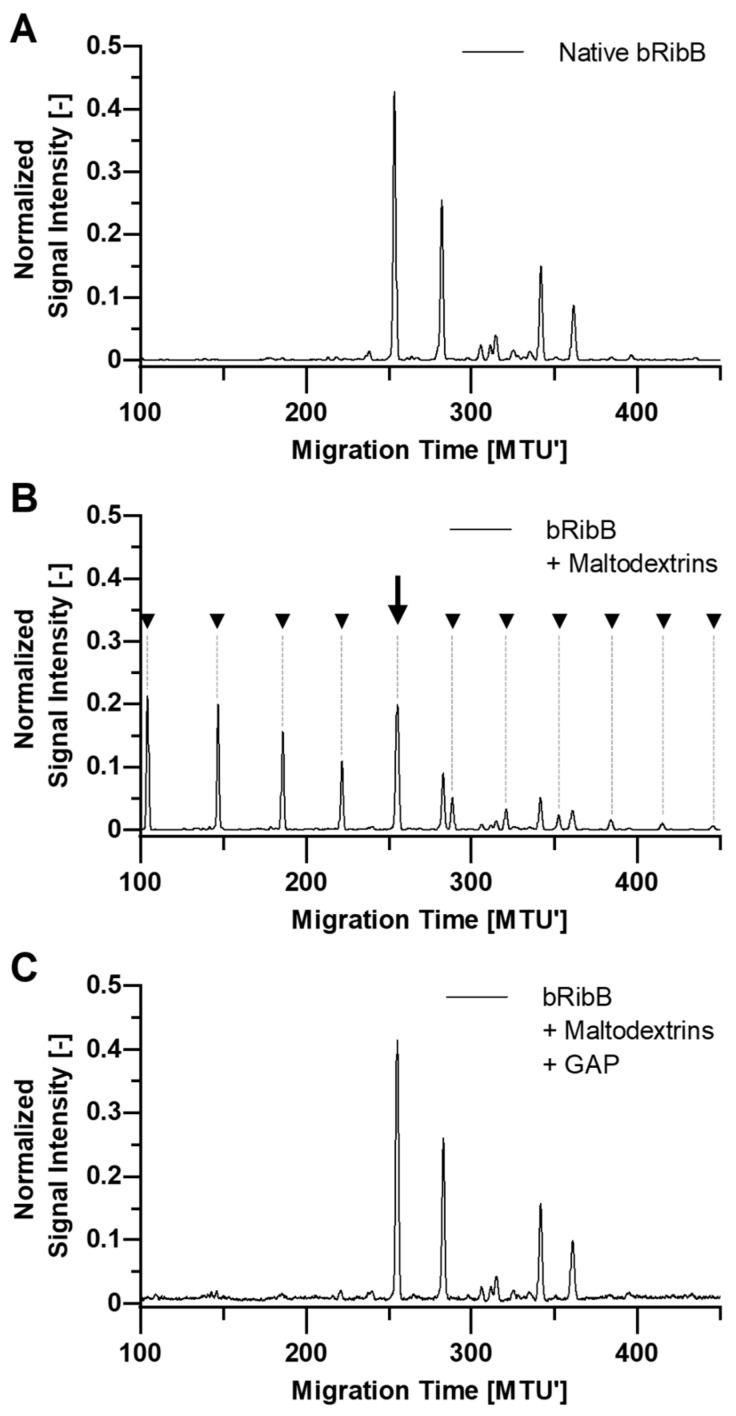
Glucoamylase P-supported degradation of spiked maltodextrins from a bovine ribonuclease B (bRibB) *N*-glycan sample. APTS-labeled *N*-glycans from bRibB (**A**) were spiked with APTS-labeled maltodextrins (**B**). Peaks of APTS-labeled maltodextrins are marked with black triangles. Treatment with glucoamylase P (GAP, 0.017 U/μL) for 30 min in disodium phosphate-citrate buffer at pH 5 restores the original *N*-glycan fingerprint (**C**). Success of maltodextrin spiking, and glucoside hydrolase treatment were controlled by xCGE-LIF. Migration time plotted on the *x*-axis was aligned to an internal migration time standard, resulting in Migration Time Units (MTU’). Signal intensity (peak height) of automatically picked peaks was normalized to a sum value of 1. Data analysis was performed using glyXtoolCE™.

**Figure 5 molecules-28-01843-f005:**
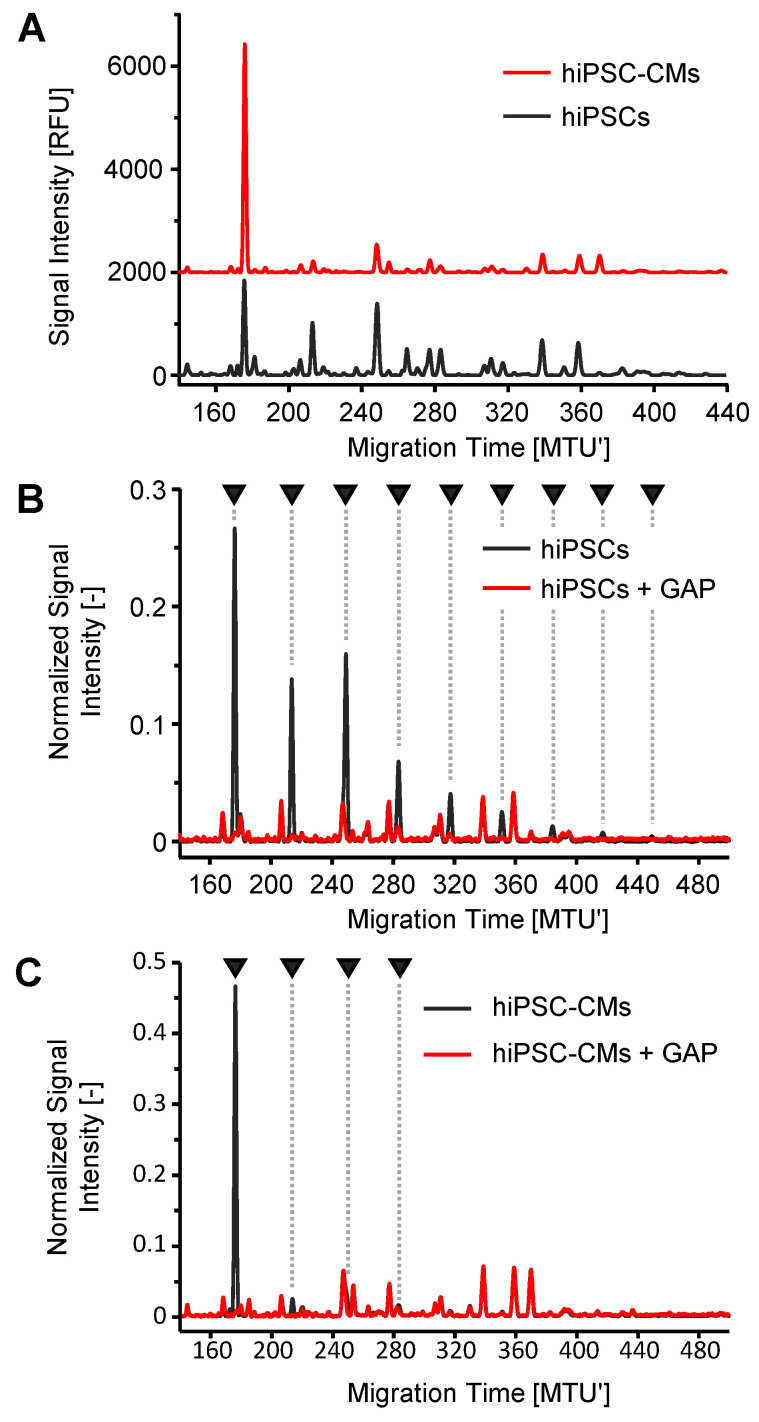
*N*-glycans of human-induced pluripotent stem cells (hiPSCs) and stem cell-derived cardiomyocytes (hiPSC-CMs) (**A**) were analyzed by xCGE-LIF without prior purification on the glycoprotein level. Interfering OSIs suspected by migration time matching to be maltodextrins in hiPSCs (**B**) and hiPSC-CMs (**C**) were confirmed and degraded by enzymatic treatment with glucoamylase P (GAP). The enzymatic degradation of interfering OSIs clarifies the differences in *N*-glycan fingerprints during cell differentiation. Peaks containing OSIs are marked with a black arrow. Signal intensity (normalized with sum of picked peak heights = 1 in **Panels B** and **C**) was plotted over aligned migration time in Migration Time Units (MTU′). For comparison of the normalized signal intensities of *N*-glycan-derived peaks, peaks of OSI degradation products were considered for signal normalization, too. Data analysis was performed using glyXtoolCE™.

**Table 1 molecules-28-01843-t001:** Effectivity of 10 commercially available glucoside hydrolases for degradation of maltodextrins and dextrans is compared by abundancies of mono- and oligosaccharides before and after treatment in disodium phosphate-citrate buffer at pH 5 for 1 h at 37 °C. Percentages of maltodextrins and dextrans were calculated after automatic peak picking with glyXtoolCE™. The most efficient enzymes for maltodextrin degradation are highlighted dark (best) and light blue (second best). The most efficient enzymes for dextran degradation are highlighted dark (best) and light green (second-best). GAP: glucoamylase P; DxChe: Dextranase from *Chaetomium erraticum*; DxPsp: Dextranase from *Penicillium* species. * GAP was used with a concentration of 0.17 U/μL.

Substrate		Maltodextrin				Dextran			
	Peak of	DP1 [%]	DP2 [%]	DP3 [%]	≥DP4 [%]	DP1 [%]	DP2 [%]	DP3 [%]	≥DP4 [%]
Enzyme									
none		1.9	5.9	13.7	78.5	0.0	0.0	1.2	98.8
ß-amylase		2.9	57.4	33.2	6.5	0.0	0.0	1.4	98.6
GAP *		0.0	85.3	14.7	0.0	1.3	62.2	0.0	36.5
Oligo-1,6-glucosidase		5.4	6.6	13.4	74.6	47.3	4.7	0.0	48.0
DxChe		0.9	10.8	73.2	15.1	0.2	60.3	29.8	9.7
DxPsp		0.0	81.4	5.7	12.9	0.0	43.9	50.0	6.1
α-amylase II-A		2.0	6.7	40.7	50.6	0.0	0.0	1.1	98.9
α-amylase IX-A		2.1	31.7	56.4	9.8	0.0	0.0	1.2	98.8
α-amylase XIII-A		2.4	29.4	60.9	7.3	0.0	0.0	1.3	98.7
α-glucosidase		46.0	0.0	0.0	54.0	0.0	0.0	1.1	98.9
Oligo-α-(1,4-1,6)-glucosidase		1.7	5.4	13.0	79.9	0.0	0.0	1.0	99.0

## Data Availability

The data that support the findings of this study are available from the corresponding author upon reasonable request.

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
