# Peer review of "Reliable N-Glycan Analysis–Removal of Frequently Occurring Oligosaccharide Impurities by Enzymatic Degradation"

_molecules, 2023, doi:10.3390/molecules28041843_

Round 1
Reviewer 1 Report
The manuscript describes development of an enzymatic procedure for purification of the oligosaccharides released from glycoproteins from glucose oligomers, often present in biological samples in high concentration, complicating further analysis. Two hydrolases were selected from 10 commercial enzymes. The treatment of labeled N-glycan samples with suitable glucoside hydrolases is advantageous to usually performed glycoprotein sample clean-up steps that are not completely removing all glucose oligomers during sample preparation. The procedure looks really useful.
I could not find errors.
Reviewer 2 Report
This is an excellent original article about a highly relevant issue that is most definitely of interest to scientists involved with glycan analysis from complex biological samples. First, the authors must be credited for raising awareness to the problem of glycan sample contamination with laddered polyhexose impurities (OSIs) and the impact of these analyte-related compounds on glycan analysis data interpretation. Second, the authors provide data that show the usefulness and applicability of several specific degrading enzymes as a suitable and practical remedy for this problem. Last but not least they show data for a comparative assessment of the proposed enzymatic OSI-destruction and different alternative approaches for OSI removal. In summary, this work will contribute to improve data quality in glycan analysis - particularly for samples directly derived from living cells and tissues.
I have a few very minor suggestions for improvement of the manuscript.
It is my personal impression that the relevance and importance of OSI contamination in glycan samples deserves a more detailed introduction and discussion. Previous publications have discussed a direct origin of OSI from the biologicals samples and test items, i.e. glycogen and cellulosic sources as the origin of OSI contamination. In the discussion however the authors present powdered gloves, cell culture media, leachables from chomatography resins and contaminations from commercial chemicals and reagents as likely sources for OSI impurities. While all these explanations all seem valid and reasonable, this rather broad view on potential OSI sources is somewhat distracting. The main question remains to discussed: Do OSI contaminations predominantly arise from the biological sample itself or are they introduced from various external sources such as cheap impure chemicals, powdered gloves etc. ? Along this train of thought it is a very interesting finding, that the authors had presented different stem cell preparations as their major test items. This is of course a very complex biological sample that involves living cells. From the bulk of literature it seems that purified individual glycoproteins have less of a problem with prominent OSI contamination. It would be nice to read some more detailed discussion as to why these glycan samples derived from stem cells that were used by te authors in their study were particularly prone to OSI contamination and could therefore serve as an ideal test item.
The resolution of most figures is fine. Only figure 2 seems to have a less than optimum resolution. To do this very nice work justice, the quality and resolution of this figure should be improved ahead of print.
The supporting figures are really relevant and helpful - and of good quality. The authors may consider some of these figures worthy to be moved into the main manuscript.
Reviewer 3 Report
This manuscript describes important issue of contamination of glycans in glycoanalytical methods and solution. This is very important. The article is well written and easy to follow, but some issues need to be addressed.
Comments.
Major
1. Reproducibility.
Table 1- were these experiments done only once? Reproducibility would be useful.
Data in supplementary figures 6, 7, 8 - it would be good to complement the data in these figures with tables including the percentage areas for the peaks to see exactly how much is the original profile affected by the oligosaccharide impurities (OSIs) and if the original peaks are not reduced significantly by the enzymes used to remove these OSIs (side reaction), do you have replicates to see reproducibility?
Minor
1. Line 59, page 2- mass spectrometry doesn’t need derivatization with chromophores (the way this is worded it implies it does). Please rewrite the sentence.
2. Line 431, page 12- size of oligosaccharides is up to a dozen (DP12), please correct this typo.
3. Original image for the supplementary figure 9- it would be useful to add the enzyme shortcuts on to the line text as they are in the text for clarity and easier to follow.
Conclusion. The manuscript if suitable for publication in Molecules after revision.
Round 2
Reviewer 3 Report
All my comments were satisfactorily answered. I have only minor comments regarding to the updated version.
Figure 3 is not cited in the Results text, it should be added there.
The updated Supplementary Figure 1 contains also HILIC-UPLC/HPLC? (UPLC in the title and HPLC in the panel G) profiles for comparison. This is very valuable, but the method for generating these profiles is missing, it should be complemented.
